# 24-Epibrassinolide-Succinic Acid Conjugate Is Involved in the Acclimation of Rape Plants to Salt Stress

**DOI:** 10.3390/plants14213404

**Published:** 2025-11-06

**Authors:** Liliya V. Kolomeichuk, Vladimir A. Khripach, Raisa P. Litvinovskaya, Aleh P. Savachka, Mingxiang Liang, Li Xu, Vladimir V. Kuznetsov, Marina V. Efimova

**Affiliations:** 1Department of Plant Physiology, Biotechnology and Bioinformatics, Biological Institute, National Research Tomsk State University, Lenin Avenue 36, Tomsk 634050, Russia; stevmv555@gmail.com; 2Institute of Bioorganic Chemistry, National Academy of Sciences of Belarus, Kuprevich Street 5/2, 220084 Minsk, Belarus; khripach@iboch.by (V.A.K.); litvin@iboch.by (R.P.L.); oleg.brsv@mail.ru (A.P.S.); 3College of Resources and Environmental Sciences, Nanjing Agricultural University, No. 1 Weigang, Nanjing 210095, China; liangmx@njau.edu.cn (M.L.); xuli602@njau.edu.cn (L.X.); 4K.A. Timiryazev Institute of Plant Physiology, Russian Academy of Sciences, Botanicheskaya Street 35, Moscow 127276, Russia; vlkuzn@mail.ru

**Keywords:** *Brassica napus* L., brassinosteroids, EBL THS conjugate, succinic acid, chloride salinization, oxidative and osmotic stress, proline, photosynthesis, enzymes, ions

## Abstract

The influence of the conjugate of 24-epibrassinolide with succinic acid (tetrahydrosuccinate of 24-epibrassinolide, EBL THS) and 24-epibrassinolide (EBL) on the acclimation of rapeseed plants (*Brassica napus* L.) to chloride salinity (150 mM NaCl) was investigated. After two weeks of growth in Hoagland–Snyder medium, the rapeseed seedlings were transferred to the same medium supplemented with EBL or EBL THS (10 nM) for 4 h, after which NaCl (150 mM) was added; parameters were taken on the 1st, 3rd, 5th and 7th days. It was established that salt stress inhibited growth processes (by 19–45%), reduced the chlorophyll and carotenoid contents (by 19–50%), photosystem II efficiency (by 13–19%), tissue hydration (by 3.54%), and osmotic potential (by three times), increased lipid peroxidation (LPO) (by 1.5–2 times), and proline accumulation (by 1.4–18 times), and altered ion status, increasing the concentrations of Na^+^ and Cl^−^ ions while decreasing the levels of K^+^, Ca^2+^, Mg^2+^, S^2+^, Fe^2+^, Al^3+^, and P^3+^. The short-term pretreatment of plants with EBL THS, similar to EBL, reduced the inhibitory effects of NaCl on growth processes, pigment content (to a greater extent with EBL THS), the efficiency of photochemical processes in photosystem II, the accumulation of Na^+^ ions, and in the case of EBL THS, the accumulation of Cl^−^ ions. Both regulators (especially EBL THS) reduced LPO, and stimulated the accumulation of NaCl-induced proline, which was organ-specific and dependent on the duration of stress. EBL THS stimulated the activity of superoxide dismutase and peroxidase, whereas EBL primarily stimulated peroxidase. Thus, it was demonstrated for the first time that EBL THS, like EBL, increased the salt tolerance of rapeseed plants, but had a more pronounced stress-protective effect, primarily at the level of antioxidant system components.

## 1. Introduction

One of the main adverse abiotic factors reducing crop yields is soil salinization [1]. Approximately 50% of irrigated land and 20% of all cultivated land are affected by excessive salinization [2]. Primary and secondary soil salinization are distinguished. Primary salinization is a consequence of natural rock weathering processes, which release soluble salts into the soil, with NaCl being the most mobile. Secondary salinization is the result of human activities, such as violations of land irrigation technology and agricultural technology for the use of mineral fertilizers [3].

Salinization initiates osmotic stress and disrupts the ionic status of plants [4], resulting in toxic effects of excess inorganic ions on cellular metabolism, the generation of reactive oxygen species (ROS), and the development of oxidative stress. Oxidative and osmotic stresses lead to a decrease in photosynthesis intensity, a decrease in productivity [5,6,7], accelerated aging processes, and even plant death [8]. Currently, 90% of all global agronomic production is achieved by cultivating 30 species of crops, the vast majority of which are glycophytes, i.e., plants that are intolerant to excessive environmental salinization [9].

Currently, there are no effective, environmentally friendly technologies to reduce land salinity. For this reason, the use of saline areas for agricultural production requires increasing the salt tolerance of economically valuable plant varieties, which involves studying the physiological mechanisms of plant adaptation to saline soils and methods for their regulation.

One strategy for enhancing plant resistance and preserving yield in saline territories is the use of growth regulators, including brassinosteroids (BRs). BRs significantly increase plant stress resistance by regulating physiological processes and cellular metabolism [10]. One of the primary mechanisms of the protective action of BRs is their ability to increase the level of cellular antioxidants, which sequester (neutralize) stress-induced reactive oxygen species [11,12,13,14]. BRs also regulate water and ionic status, thereby preventing plant wilting and reducing the toxic effects of excess inorganic ions [15]. Furthermore, BRs maintain photosynthetic activity [16,17] and contribute to preserving crop yield under stress conditions [18,19]. Exogenous BR application enhances plant resistance to heat stress and a range of other adverse environmental factors by increasing cellular antioxidant activity [20].

Currently, the creation of transformed molecules based on BRs that possess biological properties not only of BRs but also of other physiologically active substances is of significant interest. For example, conjugates based on 24-epibrassinolide and indole-3-acetic acid (IAA) have been created. The new hybrid molecule exhibited high biological activity, exceeding the activity of the initial compounds [21,22]. Moreover, conjugates with varying degrees of acylation were more effective than BRs. The physiological effects of 24-epibrassinolide (EBL) and 24-epicastasterone conjugates with salicylic acid surpassed the efficacy of BRs and salicylic acid, which act individually [23]. Conjugates of EBL with sulfuric acid exhibited adaptogenic properties in tests with chloride salinization on sunflower plants. For almost all the hybrid molecules, their protective effects on the growth of *Helianthus annuus* L. under salinization conditions were greater than those of EBL [24]. The effects of 24-epicastasterone and its conjugates with organic acids (2-monosalicylate 24-epicastasterone and 2,3,22,23-tetraindolylacetate 24-epicastasterone) on the growth and biochemical parameters of meadow clover were also studied. In vegetation experiments, 24-epicastasterone and its ester with IAA, when applied as a foliar treatment, had the greatest stimulating effect on growth processes and the content of major photosynthetic pigments in plants [25].

Conjugates of EBL and 24-epicastasterone with succinic acid are fundamentally new compounds that significantly influence the seed quality and seedling growth of spring barley [26]. These research results demonstrate the potential of using succinic acid in agronomic practice to increase plant stress resistance. In particular, succinic acid increased the activity of antioxidant enzymes and the ethylene content in wheat plants [27], exhibited antibacterial activity in tomato plants [28], and regulated stomatal conductance and increased the relative water content in kale leaves under drought conditions [29].

Previously, we established that pretreatment (for 4 h) of plants with an EBL conjugate of succinic acid (24-epibrassinolide tetrahemisuccinate, EBL THS (Figure 1)) was more effective than pretreatment with a mixture of EBL and succinic acid under stress conditions [30]. However, the question of which physiological reactions underlie the protective action of the EBL THS conjugate remains largely unanswered. Elucidating the physiological mechanisms of the stress-protective action of brassinosteroid conjugates with succinic acid holds not only scientific but also practical importance for the discovery and development of new, effective regulators of resistance and productivity in key agricultural crops.

In light of the above, the aims of our work were to investigate (1) whether the 24-epibrassinolide conjugate with succinic acid (EBL THS) enhances salt tolerance in rapeseed plants; (2) which physiological reactions are involved in the protective action of EBL THS under salt stress; and, (3) whether the specificity of the protective effect of the EBL THS conjugate differs from the protective effect of EBL alone under salinization conditions.

## 2. Results

### 2.1. Effects of EBL THS and EBL on Rapeseed Growth Under Salt Stress

The experimental data obtained showed that rapeseed plants exhibited considerable sensitivity to chloride salinization (Table 1). Starting from the first day of exposure, stem growth was suppressed, and the number of leaves and their total area decreased by 19–23% relative to those of the control plants. This ultimately led to a 40% reduction in the fresh weight of the plants. An increase in the duration of chloride salinization was expectedly accompanied by further inhibition of growth processes for all analyzed parameters, with the exception of the root system. By the end of the experiment, the hypocotyl and stem lengths in this treatment were 20% and 30% lower than those in the control, respectively, while the total leaf area and fresh plant biomass decreased by 30% relative to those in the control.

The inhibitory effect of salt stress was almost completely alleviated by preliminary treatment of plants with both EBL THS and EBL. However, a more pronounced protective effect, in terms of maintaining stem growth, leaf surface area (on the 7th day of the experiment), and fresh biomass accumulation (on the 3rd day of the experiment), was observed after the plants were treated with EBL THS. Under optimal conditions, succinic acid did not stimulate growth at low concentrations (10 and 40 nM) but accelerated growth at relatively high concentrations (0.1 and 10 mM) (Appendix A). The pretreatment of rapeseed plants for 4 h with succinic acid (10 nM) or a mixture of succinic acid and EBL (10 nM), followed by salt stress (150 mM NaCl, 7 days), also did not result in a salt-protective effect (Appendix A), unlike the EBL THS conjugate (Table 1).

### 2.2. Effect of EBL THS and EBL on the Content of Major Pigments and the Functional Activity of PSII Under Salinization

Plant productivity is directly dependent on the efficiency of photosynthesis. One of the key indicators determining the optimal functioning of a plant’s assimilatory apparatus is the content of photosynthetic pigments, which are involved not only in photosynthesis but also in its protection from ROS. Under salt stress conditions, a significant decrease in the photosynthetic pigment content was generally observed, with the most pronounced effect noted on the 5th and 7th days of the experiment, when the contents of chlorophylls (*a* and *b*) and carotenoids decreased by 40–50% relative to those of the control (Figure 2). Pretreatment of the plants with EBL THS stabilized the contents of chlorophyll *a* (on the 3rd and 5th days) and carotenoids (throughout the experiment) at the control levels. EBL partially mitigated the negative effect of the stressor on the pigment content. However, on the 7th day of salt stress, the contents of chlorophylls *a* and *b* were 23–35% lower than the control values, despite pretreatment of the plants with the EBL THS conjugate or EBL (Figure 2). Interestingly, short-term treatment of plants with succinic acid or a mixture of succinic acid and EBL under salt stress led to a slight (10–15%) increase in the content of major photosynthetic pigments (Appendix A).

To assess the functional state of the light-dependent stage of photosynthesis, the photochemical activity of photosystem 2 (PSII) was analyzed. On the first day of the experiment, the maximum quantum yield of PSII (Fv/Fm) in stressed plants was high, at 0.80. However, by the end of the experiment, it decreased by 16%, indicating a disruption in the function of the photosynthetic apparatus (Figure 3). In contrast, in the plants pretreated with EBL, THS or EBL, this parameter remained more stable throughout the experiment and, by the end of the study, did not differ from the control values. Similar trends were observed for the effective quantum yield (Y(II)), relative electron transport rate (ETR), and photochemical quenching coefficient (qP), the values of which decreased by 13–19% in plants grown under salinization conditions by the end of the experiment and did not differ substantially from those of the control in plants pretreated with EBL, THS or EBL. Concurrently, under salinization conditions, the quantum yields of nonregulated energy dissipation (Y(NO)) and regulated thermal energy dissipation (Y(NPQ)) in PSII significantly increased. Starting from day 3 until the end of the experiment, these values exceeded the control values by 1.5 and 2 times, respectively. Pretreatment of the plants with EBL, THS or EBL mitigated this effect on day 3 of the experiment, and by the end of the experiment, the values of Y(NO) and Y(NPQ) did not differ substantially from those of the control.

### 2.3. Effect of EBL THS and EBL Pretreatment on the Inorganic Ion Contents in the Leaves, Stems, and Roots of Plants Under Salt Stress

Inorganic ions play a significant role in shaping the osmotic potential of cell contents, primarily potassium ions, although sodium ions also play an important role under salinization conditions. We analyzed the concentrations of sodium, magnesium, aluminum, phosphorus, sulfur, chlorine, potassium, calcium, and iron as percentages (At%) in the leaves, stems, and roots of the plants. Compared with those in the stems, all the analyzed ions were predominant in the roots of the control plants, with the exception of potassium and magnesium ions (Table 2). As seen from the obtained data, in response to NaCl exposure, the At% of sodium and chlorine significantly increased (6–20 times relative to the control) in all parts of the plants, and the At% of potassium, calcium, magnesium, sulfur, phosphorus, aluminum (except in the stem), and iron decreased (2–4 times relative to the control). Moreover, the concentrations of chloride and sodium ions were slightly lower in the roots than in the leaves.

Compared with the salt control, pretreatment of the plants with EBL, THS or EBL, followed by salt stress, reduced the accumulation of sodium ions in the leaves and stems of the plants by 20%. EBL-induced THS also inhibited the accumulation of chlorine ions in the leaves. Both regulators partially restored the accumulation of magnesium in the stem and calcium in the leaves and stems of the plants.

### 2.4. Effects of EBL, THS and EBL on the Water and Osmotic Status of Plants Under Salt Stress

One of the main parameters reflecting a plant’s ability to withstand the osmotic effects of salt is the hydration of its tissues. The water contents in the shoots and roots of the control plants did not differ significantly (Table 3). Compared with the control, exposure to NaCl for 7 days led to a significant decrease in the LWC of rapeseed by 5%. Pretreatment of the plants with the studied regulators completely neutralized this negative effect, maintaining the water content at control levels and, in some cases, even increasing tissue hydration. The dynamics of the water content in the roots did not significantly change throughout the experiment, except on day 3, when EBL THS pretreatment followed by salt stress led to an increase in the water content, and on day 5, the water content slightly decreased (Table 3).

Under conditions of intense salt stress, reducing the osmotic potential of root cell contents is crucial for restoring their ability to absorb water from the soil solution. The osmotic potential of the leaf exudate in rapeseed under control conditions remained at approximately −0.6 MPa throughout the experiment (Figure 4). Compared with the control treatment, salt stress led to a threefold decrease in osmotic potential, starting after one day of exposure and continuing until the end of the experiment. Furthermore, pretreatment of plants with EBL THS or EBL did not affect the magnitude of leaf osmotic potential in the context of subsequent salt stress.

### 2.5. Effect of EBL THS and EBL on the Intensity of Lipid Peroxidation in Plants Under Salinization

One of the negative consequences of salinization is oxidative stress in plants, which is associated primarily with disruptions in photosynthesis and respiration. The intensity of oxidative stress in rapeseed leaves was assessed by the level of LPO (lipid peroxidation), for which the content of TBARS (thiobarbituric acid-reactive substances) in the leaves, stems, and roots of the plants served as a criterion. As shown by the presented data, in response to chloride salinization, LPO increased 1.5–2 times relative to the control in all the plant parts from the first day until the end of the experiment (Table 4). Pretreatment of the plants with EBL THS or EBL did not significantly affect the level of oxidative stress on the first day of the experiment, except for a decrease in the TBARS content in the roots and stems when the plants were subjected to EBL THS followed by salinization. On the 3rd day of salt exposure, pretreatment of plants with EBL, THS or EBL reduced the intensity of LPO in the roots but not in the leaves or stems. On the 7th day of salt exposure, pretreatment of plants with EBL or EBL THS completely abolished NaCl-dependent oxidative stress in the leaves, and moreover, EBL reduced the level of oxidative stress in the roots.

### 2.6. Effect of EBL THS and EBL on Proline Content in Plants Under Chloride Salinization

Compatible osmolytes with chemical chaperone properties play an important role in plant adaptation to water deficit and the toxic effects of excess inorganic ions, primarily sodium. The plants responded to salt stress via intense proline accumulation in all the plant parts, starting from the first day of the experiment (Figure 5). The peak accumulation of the compatible osmolyte was observed on the 5th day of salt stress in the leaves and on the 7th day in the stems and roots, with the proline content exceeding the control level by 10–18 times. Preliminary treatment of plants with the studied regulators (to a greater extent with EBL THS) increased the proline accumulation induced by NaCl; however, this effect was organ-specific and depended on the duration of stress exposure. Thus, EBL pretreatment promoted more intense proline accumulation in leaves (by 34%), whereas EBL THS activated imino acid accumulation in stems and roots (3.1- and 4.9 fold, respectively). On the third day of the experiment, more intense proline accumulation was observed only in the roots and stems in the EBL THS treatment. On the fifth day, EBL THS stimulated the accumulation of NaCl-induced proline in stems and roots by 2.1–2.2 times, whereas EBL did so by 1.0–1.9 times. By the end of the experiment, EBL and EBL THS increased proline accumulation in stems (by 2.0 and 2.3 times, respectively) and in roots (by 1.2 and 2.0 times, respectively). In addition, EBL-induced THS also increased the proline content in leaves by 1.8 fold.

### 2.7. Effects of EBL, THS and EBL on the Activity of Superoxide Dismutase and Peroxidase in Rapeseed Leaves Under Salt Stress

To prevent the development of oxidative stress in plants, cellular antioxidant systems capable of quenching reactive oxygen species are activated. In the present study, the activity of superoxide dismutase in plants pretreated with EBL THS or EBL followed by salt stress did not significantly differ from that in the control (Figure 6). The only exception was the treatment with EBL THS followed by salt stress (“EBL THS + NaCl”), where the enzyme activity exceeded the control values by 3 and 0.3 times on the first and seventh days of the experiment, respectively.

Salt stress did not cause an increase in peroxidase activity in rapeseed leaves; however, compared with the control, the hormonal pretreatment of plants followed by salt stress stimulated enzyme activity (Figure 7).

The treatment of plants with EBL increased peroxidase activity by 1.7, 2.7, and 1.8 times on the 1st, 5th, and 7th days of salt exposure, respectively. The treatment of plants with EBL THS increased peroxidase activity by 1.7- and 1.8 fold on the 1st and 5th days of salt stress, respectively. Importantly, compared with the control treatment, the EBL and EBL THS treatments increased peroxidase activity not only in relation to the control but also in relation to salt stress (Figure 7).

## 3. Discussion

### 3.1. Pretreatment of Rapeseed with EBL THS and EBL Regulates Growth and Osmotic Status of Plants Under Salt Stress

A common plant response to salinity is rapid inhibition of growth processes. This is due to a decrease in the water absorption capacity of the root system and the development of water deficit, which leads to metabolic disruption and reduced productivity [31]. In our study, chloride salinization likely inhibited plant growth. Significant suppression of stem and leaf growth, a reduction in biomass accumulation, and a decrease in tissue hydration were observed (Table 1 and Table 3). This effect of salt stress was accompanied by a decrease in the osmotic potential of the leaf cell contents and alterations in ionic homeostasis (Figure 4, Table 2). In accordance with the understanding of the key role of K^+^ and Na^+^ in plant acclimatization to osmotic stress [32], we observed active accumulation of sodium and chlorine ions with a simultaneous decrease in the concentrations of other studied ions under salinization. Such an imbalance in ionic composition exacerbates the negative effect of osmotic stress, causing toxicity from excessive sodium ions and disrupting cellular metabolism.

Brassinosteroids (BRs) increase plant resistance to osmotic stress [33], partly by improving water use efficiency and reducing stomatal conductance [34]. Our findings revealed that, compared with EBL, pretreatment of plants with EBL THS was accompanied by a more pronounced reduction in the inhibitory effects of salt stress on growth processes and tissue hydration, as well as on the inhibition of sodium and chlorine ion uptake (Table 1, Table 2 and Table 3). The observed protective effect of the EBL THS conjugate is evidently due to the realization of biological activity primarily from the brassinosteroid component of the hybrid molecule, rather than its succinic acid residue, as succinic acid at low concentrations did not stimulate growth processes under optimal conditions and did not reduce the inhibitory effect of salt stress (Appendix A). The differences in the action of the mechanical mixture of EBL and SA and the EBL THS conjugate may be related to the gradual release of free BRs and SA as a result of conjugate hydrolysis, which can be considered a form of depot.

### 3.2. EBL THS and EBL Reduce the Inhibitory Effects of Salt Stress on the Contents of Chlorophylls and Carotenoids and the Functional Activity of PSII

Plant productivity is determined by photosynthetic activity, which directly depends on the contents of major photosynthetic pigments [35]. In the present study, the high sensitivity of the rapeseed photosynthetic apparatus to NaCl exposure was demonstrated. Similar effects have also been reported, for example, in rice [36] and potato plants [37]. A rapid decrease in the photosynthetic pigment content (Figure 2) and changes in the photochemical process parameters (Figure 3) in response to salt stress indicate impaired PSII function and reduced photosynthetic efficiency. Moreover, NaCl inhibits primary photosynthetic processes in potato plants by reducing the electron transport rate and the maximum and effective quantum yields of PSII, which may be related to the toxic effects of sodium and chlorine ions, leading to the disruption of plastoquinone pool oxidation [38]. Short-term pretreatment of rapeseed plants with EBL THS significantly alleviated the negative impact of salt stress on photosynthetic processes (Figure 2 and Figure 3). Specifically, analysis of the results revealed that the EBL THS application more effectively maintained the chlorophyll and carotenoid contents in rapeseed leaves than did EBL, especially during the early stages of salt stress (Figure 2). Notably, preliminary treatment of plants with EBL-induced THS not only maintained the pigment content under salt stress but also restored the magnesium content in the leaves (Table 3), which is directly linked to increased efficiency of chloroplast function, as magnesium is the central atom in the chlorophyll molecule. This finding highlights the potential advantages of the EBL THS conjugate, compared with EBL, in protecting the photosynthetic apparatus under salinization, as it ensures the preservation of carotenoids at control levels after 7 days of salt stress (Figure 2). These findings suggest that EBL THS may have a more pronounced antioxidant effect and/or ability to maintain the structure and function of PSII under salt stress conditions.

### 3.3. EBL THS Is More Effective than EBL in Increasing the Antioxidant Activity of Rapeseed Plants Under Salt Stress

Under optimal plant growth conditions, there is a balance between the systems of ROS generation and inactivation, which is disrupted under salt stress. It is now well known that in response to intense salinity, osmotic stress develops in plants, and high concentrations of inorganic ions accumulate. This leads to disruption of cellular metabolism and inhibition of integral physiological processes, which is accompanied by the generation of ROS and the development of oxidative stress. The development of oxidative stress is based on a decrease in carbon dioxide entry into cells due to a decrease in stomatal permeability and an increase in the excitation energy of electrons [39]. In addition to the inhibition of photosynthesis, salt stress also inhibits respiration, which also leads to the generation of ROS. A significant increase in ROS levels causes the degradation of nucleic acids, protein denaturation, and membrane lipid peroxidation. ROS-dependent damage to cell membranes, including chloroplast and mitochondrial membranes, is one of the main negative manifestations of oxidative stress [40]. A commonly accepted method for assessing the intensity of oxidative stress and the structural integrity of membranes is the assessment of the content of substances that interact with thiobarbituric acid (TBARS) [41].

In our study, organ-specific accumulation of TBARS was established in rapeseed plants under salt stress (Table 4). Exposure to NaCl led to an increase in the TBARS content in all investigated organs, indicating the development of oxidative stress. Starting from the 3rd day of NaCl exposure, pretreatment of plants with EBL THS, and to a lesser extent EBL, significantly reduced the level of lipid peroxidation, as evidenced by the decrease in TBARS levels in the leaves, stems, and roots of the plants (Table 4). Importantly, pretreatment with EBL THS almost completely eliminated the NaCl-induced increase in lipid peroxidation levels on the 7th day of the experiment, whereas EBL treatment only partially reduced the intensity of lipid peroxidation (Table 4).

Plants respond to oxidative stress by increasing the activity of the cellular antioxidant system, which ensures the sequestration of ROS. This defense system includes antioxidant enzymes (SOD, PO, catalase, etc.) and low-molecular-weight organic compounds that exhibit antioxidant properties (photosynthetic pigments, flavonoids, nonstructural carbohydrates, amino acids, etc.) [42]. A universal plant response to salt stress is the intensive accumulation of proline. The level of stress-induced proline depends on the intensity and duration of excess salinity, as well as on the species, genotype, and stage of ontogenetic development of the plant. Proline, which is a “chemical chaperone” and a nonenzymatic antioxidant, protects components of the chloroplast electron transport chain under stress, maintains enzymes, such as RuBisCO, in a functionally active state, and reduces membrane lipid peroxidation. Importantly, proline, as we have shown previously, accumulates intensively in all studied plant parts in response to the action of NaCl [43].

Pretreatment of plants with the tested growth regulators (to a greater extent, EBL THS) stimulated NaCl-induced proline accumulation, which confirms their role in enhancing antioxidant defense and osmoregulation (Figure 5). The observed effect was organ-specific and dependent on the duration of stress exposure, indicating the complex nature of proline metabolism regulation under the influence of brassinosteroids and their derivatives with succinic acid.

Importantly, proline has properties not only as a “chemical chaperone” but also as a nonenzymatic antioxidant, which is important for understanding the protective role of EBL THS under salt stress. Proline reduces the level of membrane lipid peroxidation [44] and sequesters hydroxyl radicals, the most aggressive ROS [45]. Rehman, A.U. et al. [46] demonstrated that proline also quenches singlet oxygen and superoxide radicals (O_2_^•−^) in vitro through an electron transfer mechanism. In addition to direct ROS inactivation, proline can induce the expression of antioxidant enzyme-encoding genes and stimulate their activity under salt stress, which may be mediated by its protective effect as a “chemical chaperone” [44,47,48].

Furthermore, a differential response of antioxidant enzymes to short-term pretreatment of plants with EBL THS and EBL followed by chloride salinization was demonstrated. The plants treated with EBL THS presented increased SOD (Figure 6) and PO (Figure 7) activities, whereas exogenous EBL stimulated only PO activity (Figure 7). These findings indicate that EBL, THS and EBL have different influences on the regulation of antioxidant enzymes in response to salt stress. Importantly, the increase in PO activity under the influence of EBL THS also correlates with less lipid damage, as evidenced by the reduced content of TBARS (Table 4), which may indicate more effective membrane protection when EBL THS is used. Similar data were obtained earlier on wheat coleoptiles under high-temperature stress [49], which suggests the universality of the stress-protective effect of brassinosteroids under the damaging action of abiotic factors of different physical natures.

## 4. Materials and Methods

### 4.1. Plant Materials and Experimental Design

24-Epibrassinolide (EBL) and its 2,3,22,23-tetrahemisuccinate (EBL THS) were synthesized at the Laboratory of Steroid Chemistry, Institute of Bioorganic Chemistry, National Academy of Sciences of Belarus. EBL THS is a fairly stable compound. This is confirmed by the fact that storing EBL THS at −5 °C for one year does not alter its spectral or chromatographic characteristics (Appendix A).

Rapeseed (*Brassica napus* L.), an important agricultural and technical crop, was chosen as the object of study. Rapeseed seeds were germinated on perlite for 7 days, after which the seedlings were transferred to hydroponic conditions on a liquid Hoagland–Snyder nutrient solution (HS) under L36W/77 Fluora fluorescent lamps (“Osram”, Munich, Germany) with a photosynthetic photon flux density (PPFD) of 200–250 μM m^−2^ s^−1^ in a phytotron with a 16 h photoperiod and a temperature of 20 ± 3 °C. After two weeks of growth in the hydroponic system, the plants were transferred to the same medium supplemented with EBL THS or EBL at a concentration of 10 nM for 4 h. The plants were subsequently placed on HS nutrient media (without steroid compounds) supplemented with 0 or 150 mM NaCl. In addition, other groups of plants were treated (4 h) with different concentrations of succinic acid and grown under control conditions for 7 days. A portion of the plants were pretreated with succinic acid or a mixture of succinic acid and EBL (10 nM) and grown for 7 days under salt stress conditions. A mixture of EBL with succinic acid was used as a control. According to our observations, individual components of the ester (in our case, a steroid alcohol and an organic acid) are capable of producing a physiological effect in plant cells similar to that of the ester itself (in our case, tetrasuccinate). One explanation for this may be the possibility of ester hydrolysis by cellular enzymes and the independent role of its chemical components as active factors. In this case, their separate entry into the cell (even taking into account altered bioavailability) can reproduce the effect of the ester, which is of practical interest. The concentrations of EBL THS, EBL, and NaCl used in this work were selected in preliminary experiments. The plant material was typically harvested on the 1st, 3rd, 5th, and 7th days after the addition of NaCl to the medium.

### 4.2. Determination of Plant Biomass

The plant biomass was determined gravimetrically. For fresh weight estimation, an analytical balance with an accuracy of 1 mg (OHAUS RV-313 (Parsippany, NJ, USA)) was used, whereas for dry weight estimation, an analytical balance with an accuracy of 0.1 mg (AB54-S, Mettler Toledo, Greifensee, Switzerland) was used. In the latter case, the plant samples were predried to a constant weight at 70 °C.

### 4.3. Determining the Osmotic Potential

The osmotic potential of the cell exudates was determined via an Osmomat 030 cryoscopic osmometer (Gonotec, Berlin, Germany) according to the manufacturer’s instructions. The cell sap was squeezed from defrosted leaf samples.

### 4.4. Determination of Photosynthetic Pigments

The Lichtenthaler method was used to estimate the contents of chlorophyll *a* (Chl *a*), chlorophyll *b* (Chl *b*), and carotenoids (Car) [50]. For this purpose, leaf samples (70 mg) were ground in liquid nitrogen, after which the resulting material was transferred to a tube with 96% ethanol (1.5 mL) and calcium carbonate and vigorously vortexed, after which the homogenate was centrifuged for 10 min at 8000× *g* via a MiniSpin centrifuge (Eppendorf, Hamburg, Germany). This extraction was performed three times. The extracts were combined, and the total volume was 5 mL. To estimate the concentration of pigments in the alcoholic extract, the Lichtenthaler formula was used after the optical density of the extract was measured at wavelengths of 470, 648, 664, and 720 nm on a Genesys 10S UV–Vis spectrophotometer.

### 4.5. Determination of Chlorophyll Fluorescence

The photochemical activity parameters of PS II were measured via an RAM fluorimeter (MINI-PAM-II, Heinz-Walz, Effeltrich, Germany). The fluorescence coefficients and relative electron transport rates were calculated via MINI-PAM-II software WinControl-3 (https://www.walz.com/downloads/products accessed on 25 February 2025). The parameters evaluated were Fv/Fm (maximum quantum efficiency of photosystem II, PS II), ETR (electron transport rate), Y(II) (photochemical utilization of excitation energy in photosystem II under active light), qP (photochemical fluorescence quenching coefficient), and NPQ (nonphotochemical fluorescence quenching).

### 4.6. Evaluating Lipid Peroxidation Levels

The level of lipid peroxidation was evaluated by the formation of a stained complex between thiobarbituric acid and thiobarbituric acid-reactive substances (TBARS) upon heating according to Buege and Aust [46,51]. Fresh leaf samples (100 mg) were homogenized in 1 mL of 10% trichloroacetic acid (TCA). To 0.5 mL of this homogenate, 1.5 mL of 0.8% TBA solution was added. The samples were incubated in a water bath at 5 °C for 30 min, followed by cooling on ice. After centrifugation at 10,000 rpm for 10 min to clear the precipitate, the absorbance of the supernatants was measured at 532 nm and 600 nm via a Genesys 10S UV–Vis spectrophotometer (Thermo Fisher Scientific, Waltham, MA, USA).

### 4.7. Determination of the Free Proline Content

The free proline content in the plant tissues was determined according to the methods of Bates et al. [47,52] with minor modifications. Proline, a key imino acid involved in osmotic regulation and protection of plant cells against stress, was measured spectrophotometrically.

For analysis, 50 mg of fresh leaf tissue was used. Leaf samples were ground with 1 mL of 3% sulfosalicylic acid (SSA). The resulting suspension was mixed for 5 min on a shaker. The suspension was subsequently centrifuged at 10,000 rpm for 15 min at 4 °C. To 0.5 mL of the supernatant, 0.5 mL of 96% trichloroacetic acid (TCA) and 0.5 mL of 2.5% ninhydrin solution were added. The mixture in the test tubes was mixed and incubated in a water bath at 95 °C for 60 min. After incubation, the tubes were cooled on ice. Then, 2 mL of toluene was added, and the mixture was vigorously shaken for 15–20 s to extract the colored compound (the product of the proline–ninhydrin reaction). The upper toluene layer, containing the colored product, was carefully separated. The absorbance of the toluene extract was measured via a Genesys 10S UV–Vis spectrophotometer (Thermo Fisher Scientific, Waltham, MA, USA) at a wavelength of 520 nm. The free proline content was calculated via a calibration curve constructed with standard solutions of L-proline. The results are expressed as μmol proline per gram of fresh weight (μmol/g FW).

### 4.8. Determination of the Activity of Antioxidant Enzymes

The total superoxide dismutase (EC 1.15.1.1.1) and guaiacol-dependent peroxidase (EC 1.11.1.7) activities were determined in fresh extracts of the leaf tissues. Leaf samples (200 mg) were ground in liquid nitrogen with insoluble polyvinyl pyrrolidone, extracted in 0.066 M potassium–phosphate buffer (pH 7.4) containing 0.5 mM dithiothreitol (Sigma–Aldrich, Darmstadt, Germany) and 0.1 mM phenylmethylsulfonyl fluoride in dimethyl sulfoxide (Sigma–Aldrich, Germany) and then centrifuged for 20 min at 8000 rpm and 4 °C via a 5430R centrifuge (Eppendorf, Hamburg, Germany). The total SOD activity was determined according to the methods of Beauchamp and Fridovich [53]. The reaction medium (2 mL) contained 10 µL of supernatant, 1.75 mL of 50 mM Tris-HCl buffer (pH 7.8), 0.2 mL of 0.1 M DL-methionine (Sigma–Aldrich, Steinheim, Germany), 0.063 mL of 1.7 mM nitro blue tetrazolium (Fermentas, Waltham, MA, USA), 0.047 mL of 1% Triton X-100 (Sigma–Aldrich, Steinheim, Germany), and 0.060 mL of 0.004% riboflavin (Sigma–Aldrich, Germany). The reaction proceeded under LED lamps (I = 232 µmol photons/m^−2^ s^−1^) for 30 min. The absorption was measured at 560 nm via a Genesys 10S UV–Vis spectrophotometer (Thermo Fisher Scientific, Waltham, MA, USA).

Guaiacol-dependent peroxidase activity was determined as previously described [54]. The reaction mixture contained 50 µL of supernatant, 1.95 mL of 0.066 M potassium-phosphate buffer (pH 7.4), 200 µL of 7 mM guaiacol (Sigma–Aldrich, Steinheim, Germany), and 250 µL of 0.01 M H_2_O_2_. The absorption was measured at 470 nm via a Genesys 10S UV–Vis spectrophotometer (Thermo Fisher Scientific, Waltham, MA, USA).

### 4.9. Determination of Total Protein Content

The Esen method was used to determine the protein content of the plant material [55]. This method is based on the staining of protein on filter paper with Coomassie Brilliant Blue G-250 dye 50. Five microliters (5 µL) of extract were applied to filter paper, dried, and then stained with a Coomassie dye solution. After drying, the stained protein was eluted in 3 mL of 1% SDS solution, after which the absorbance of the eluate was measured spectrophotometrically at 600 nm.

### 4.10. Statistical Analysis

Each experiment was repeated at least three times. The number of plants per biological replicate ranged from nine to twelve. The values are expressed as the means ± SDs. The results were evaluated via one-way analysis of variance (ANOVA) followed by Duncan’s multiple range test (DMRT) via the SPSS 26 software package. *p* values less than 0.05 were regarded as statistically significant.

## 5. Conclusions

Our results convincingly demonstrate that short-term pretreatment of rapeseed plants with an EBL-succinic acid conjugate (EBL THS) or EBL leads to increased resistance to salt stress. This is manifested in the ability of EBL THS, and to a lesser extent, EBL, to almost completely alleviate the inhibitory effects of NaCl on stem growth, leaf area (on the 7th day of the experiment), and fresh biomass accumulation (on the 3rd day of the experiment). The protective action of the studied regulators is based on various physiological mechanisms. In particular, chloride salinization disrupts ionic homeostasis and results in the accumulation of toxic sodium and chloride ions. Treatment of plants with EBL THS inhibits the uptake of sodium and chloride ions into the plant, whereas EBL only inhibits the accumulation of sodium ions. This reduces the content of toxic ions and their negative impact on cellular metabolism. The osmotic and toxic effects of salt are accompanied by the generation of ROS and the development of oxidative stress. EBL THS and EBL reduced the intensity of lipid peroxidation in roots and partially in leaves and stems (on the 3rd day of salt stress) and completely alleviated NaCl-induced lipid peroxidation in leaves on the 7th day of the experiment, whereas EBL THS reduced oxidative stress in roots and stems as early as the 1st day of salt exposure. In response to ROS generation during salinization, plants exhibit increased antioxidant activity. Pretreatment of rapeseed plants with EBL THS was accompanied by an increase in the activities of both peroxidase and superoxide dismutase, whereas EBL stimulated only peroxidase activity. Furthermore, both regulators stimulated NaCl-induced proline accumulation, which has pronounced antioxidant and osmoregulatory effects, as well as chemical chaperone properties. Compared with the control treatment, the treatment of plants with steroid preparations did not affect the osmotic potential of plant cell exudates under salt stress but significantly relieved the water deficit caused by salt stress; in some cases, EBL THS increased tissue hydration. Finally, both studied regulators had a protective effect on photosystem II (PSII) function. These compounds partially alleviated the negative impact of salt stress on the contents of major photosynthesis-related enzymes and restored Mg^2+^ accumulation, which is crucial for chlorophyll synthesis. Pretreatment of the plants with EBL THS stabilized the contents of chlorophyll a (on the 3rd and 5th experimental days) and carotenoids throughout the entire experiment at the control levels.

Thus, EBL THS more effectively than EBL enhances the resistance of rapeseed plants to salt stress by activating antioxidant systems, regulating water status and ionic homeostasis, leading to intensive proline accumulation, and maintaining the functional activity of PSII and the level of major photosynthetic pigments.

## Figures and Tables

**Figure 1 plants-14-03404-f001:**
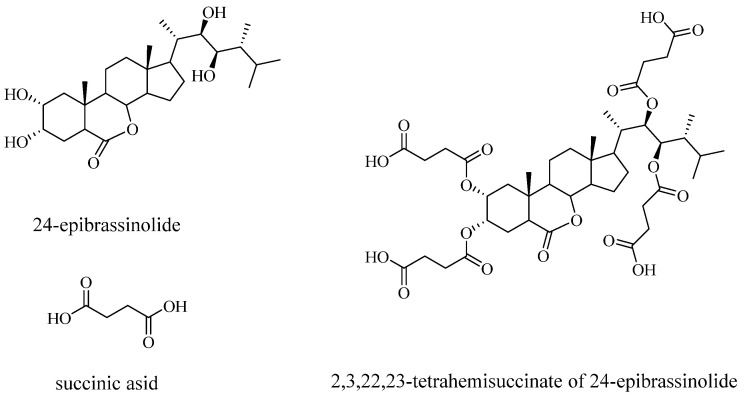
Chemical structure of 24-epibrassinolide and its conjugate.

**Figure 2 plants-14-03404-f002:**
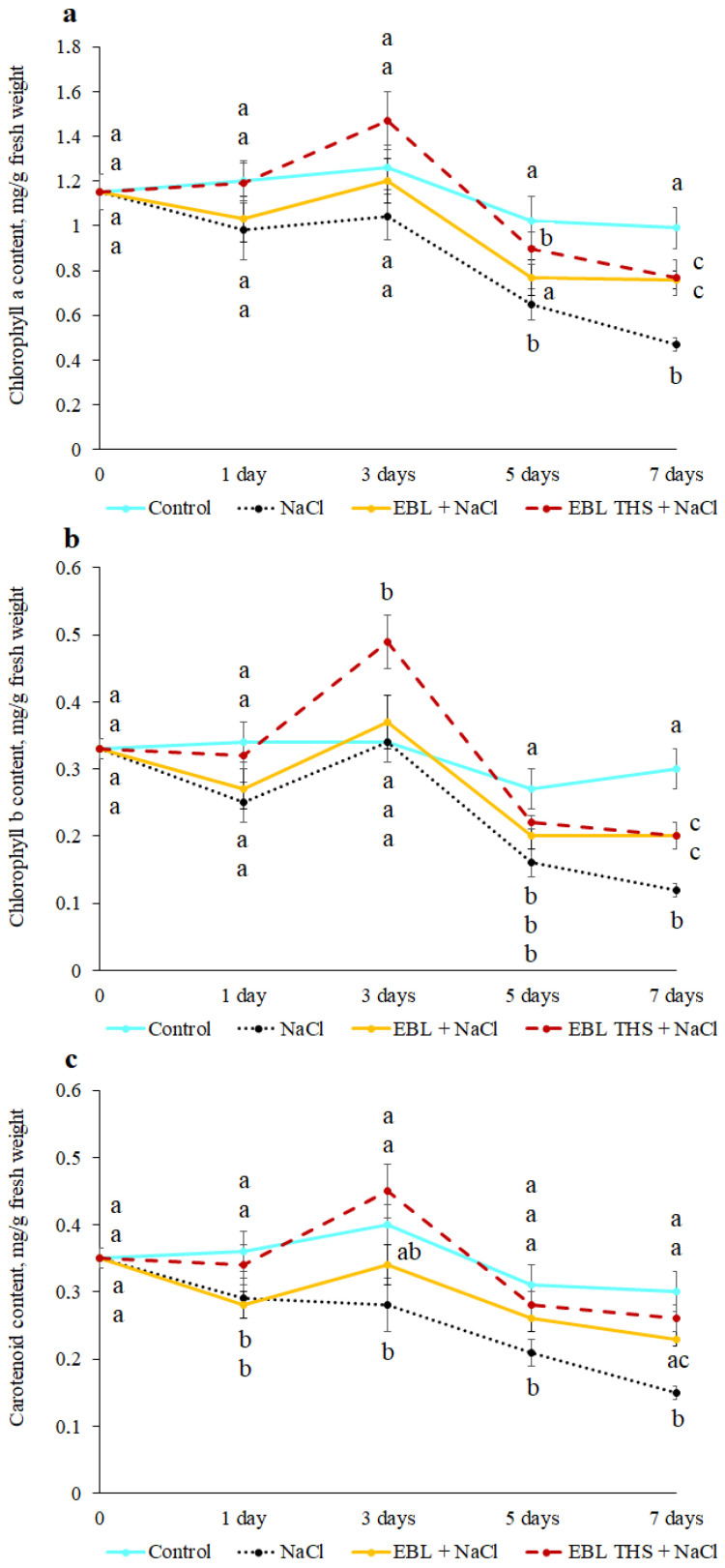
Dynamics of major photosynthetic pigment contents in rapeseed plants (*Brassica napus* L.) in response to pretreatment (4 h) with 24-epibrassinolide (EBL) or 24-epibrassinolide tetrahemisuccinate (EBL THS) followed by salt stress. (**a**)—chlorophyll *a* content; (**b**)—chlorophyll *b* content; (**c**)—carotenoid content. The values are presented as the means ± SDs for each treatment. Values not sharing a common or same alphabet letter (a–c) differ significantly at *p* < 0.05 (Duncan’s multiple range test).

**Figure 3 plants-14-03404-f003:**
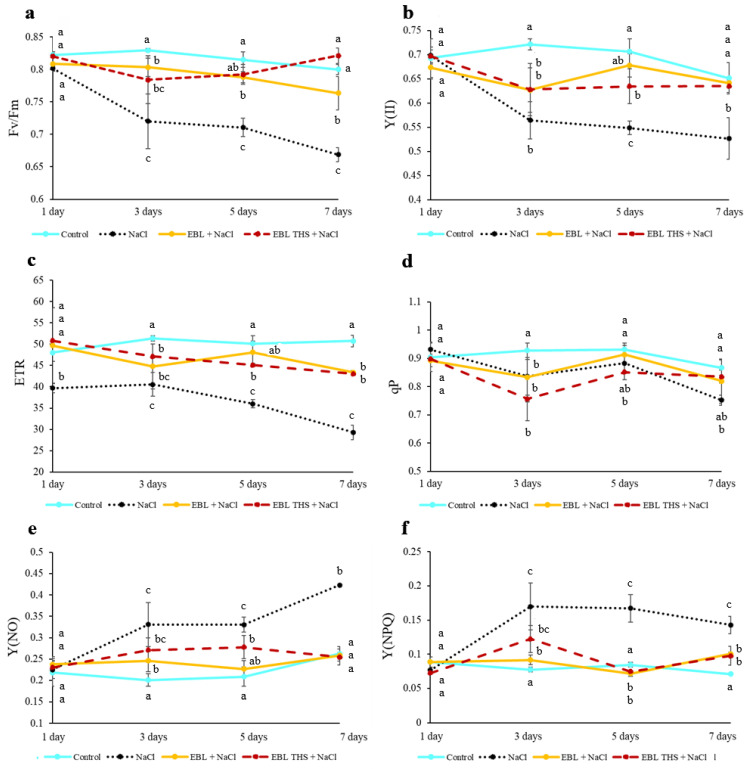
Effect of EBL THS and EBL pretreatment on the dynamics of PSII fluorescence parameters under salt stress. (**a**)—maximum quantum yield; (**b**)—effective quantum yield; (**c**)—relative electron transport rate; (**d**)—photochemical quenching coefficient; (**e**)—quantum yield of nonregulated light energy dissipation (**f**)—quantum yield of regulated thermal energy dissipation. Values not sharing a common or same alphabet letter (a–c), differ significantly at *p* < 0.05 (Duncan’s multiple range test).

**Figure 4 plants-14-03404-f004:**
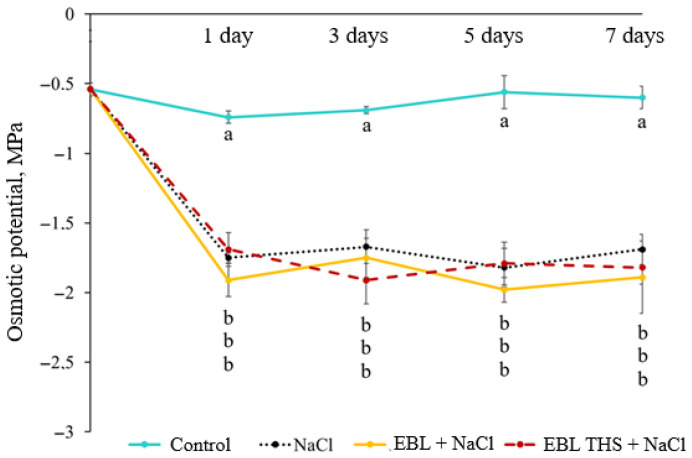
Dynamics of the osmotic potential of the cell content in rapeseed plant (*Brassica napus* L.) leaves in response to pretreatment (4 h) with 24-epibrassinolide tetrahemisuccinate (EBL THS) and 24-epibrassinolide (EBL) followed by salt stress. Values not sharing a common or the same alphabet letter (a, b) differ significantly at *p* < 0.05 (Duncan’s multiple range test).

**Figure 5 plants-14-03404-f005:**
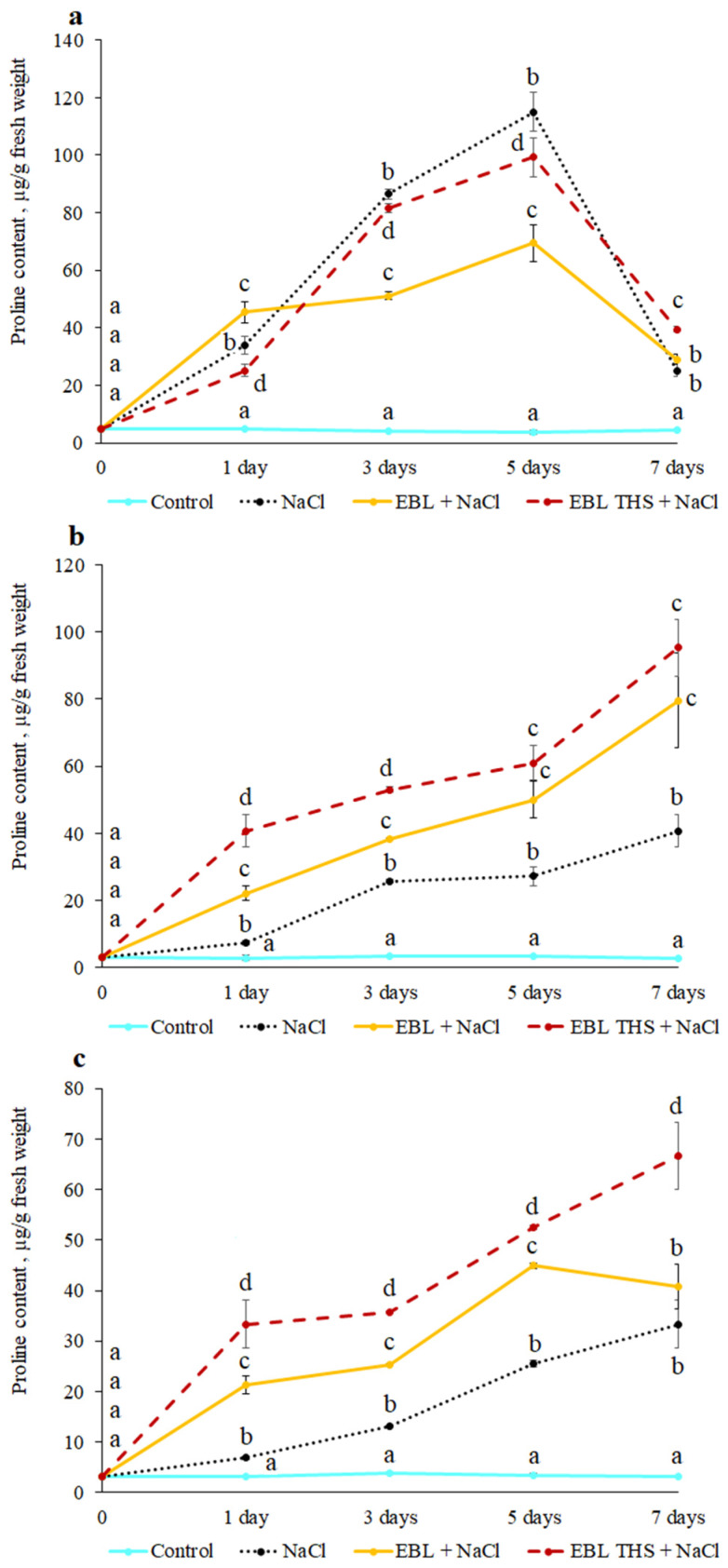
Dynamics of proline accumulation in rapeseed plants (*Brassica napus* L.) in response to pretreatment (4 h) with 24-epibrassinolide tetrahemisuccinate (EBL THS) and 24-epibrassinolide (EBL) followed by salt stress. (**a**)—leaves, (**b**)—stem, (**c**)—roots. The values are presented as the means ± SDs for each treatment. Values not sharing a common or same alphabet letter (a–d) differ significantly at *p* < 0.05 (Duncan’s multiple range test).

**Figure 6 plants-14-03404-f006:**
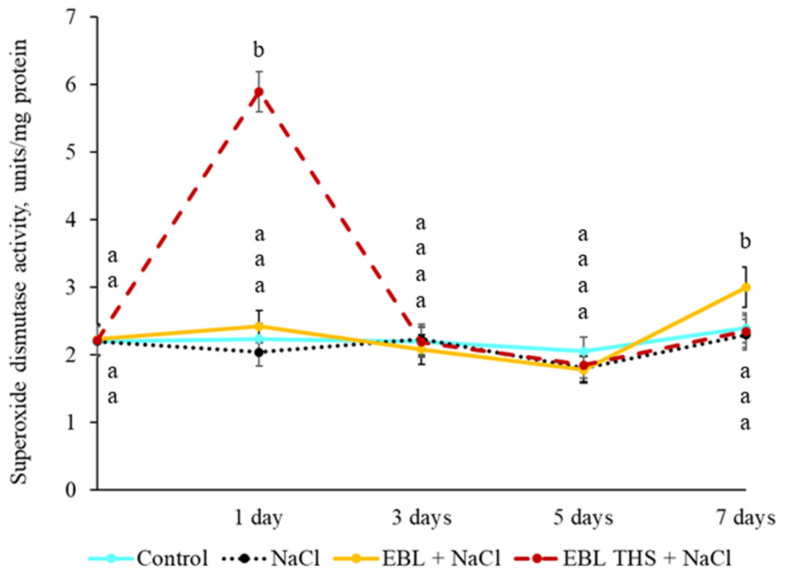
Dynamics of superoxide dismutase activity in rapeseed plants (*Brassica napus* L.) in response to pretreatment (4 h) with 24-epibrassinolide tetrahemisuccinate (EBL THS) and 24-epibrassinolide (EBL) followed by salt stress. The values are presented as the means ± SDs for each treatment. Values not sharing a common or same alphabet letter (a, b) differ significantly at *p* < 0.05 (Duncan’s multiple range test).

**Figure 7 plants-14-03404-f007:**
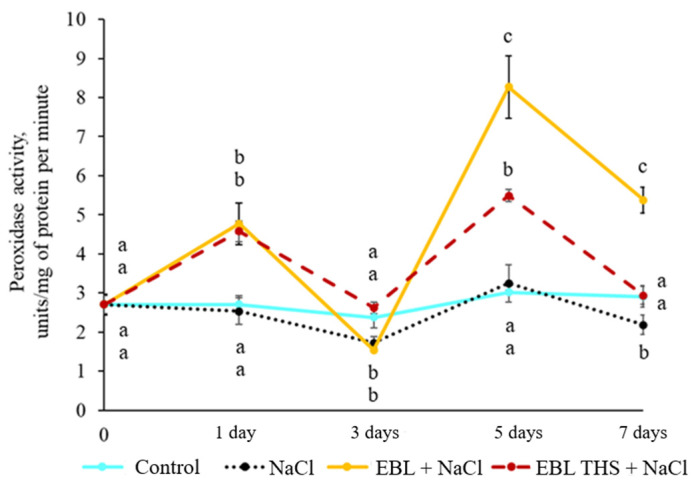
Dynamics of peroxidase activity in rapeseed plants in response to pretreatment (4 h) with 24-epibrassinolide (EBL) or 24-epibrassinolide tetrahemisuccinate (EBL THS) followed by salt stress. Values not sharing a common or the same alphabet letter (a–c) differ significantly at *p* < 0.05 (Duncan’s multiple range test).

**Table 1 plants-14-03404-t001:** Dynamics of rapeseed plant (*Brassica napus* L.) growth parameters in response to preliminary treatment (4 h) of plants with 24-epibrassinolide (EBL) or 24-epibrassinolide tetrahemisuccinate (EBL THS) followed by salt stress.

Treatment		Hypocotyl Length (cm)	Stem Length (cm)	Root Length (cm)	Number of Leaves	Leaf Surface Area (cm^2^)	Fresh Weight of Entire Plant (g)
1 day	Control	3.33 ± 0.21 ^a^	3.28 ± 0.12 ^a^	17.20 ± 0.88 ^a^	4.50 ± 0.23 ^a^	78.46 ± 5.58 ^a^	4.21 ± 0.35 ^a^
	NaCl	3.25 ± 0.25 ^a^	2.66 ± 0.12 ^b^	17.21 ± 0.31 ^a^	3.50 ± 0.25 ^b^	61.28 ± 7.57 ^ab^	2.59 ± 0.30 ^b^
	EBL + NaCl	3.46 ± 0.12 ^a^	2.90 ± 0.23 ^a^	17.10 ± 0.42 ^a^	3.80 ± 0.36 ^b^	80.22 ± 8.45 ^a^	3.21 ± 0.24 ^c^
	EBL THS + NaCl	4.18 ± 0.34 ^b^	3.05 ± 0.19 ^a^	18.10 ± 2.54 ^a^	3.80 ± 0.36 ^b^	69.64 ± 5.32 ^a^	2.49 ± 0.27 ^b^
3 days	Control	3.55 ± 0.25 ^a^	3.40 ± 0.18 ^a^	17.90 ± 1.30 ^a^	4.90 ± 0.23 ^a^	88.90 ± 6.01 ^a^	4.44 ± 0.40 ^a^
	NaCl	3.50 ± 0.26 ^a^	2.80 ± 0.20 ^ab^	17.95 ± 1.80 ^a^	3.50 ± 0.26 ^b^	65.98 ± 2.70 ^b^	2.94 ± 0.28 ^b^
	EBL + NaCl	3.55 ± 0.32 ^a^	3.05 ± 0.16 ^a^	17.80 ± 1.94 ^a^	3.94 ± 0.32 ^b^	82.06 ± 8.89 ^a^	3.25 ± 0.39 ^b^
	EBL THS + NaCl	4.25 ± 0.46 ^a^	3.11 ± 0.29 ^a^	18.64 ± 0.25 ^a^	3.89 ± 0.27 ^b^	89.59 ± 6.43 ^a^	3.81 ± 0.54 ^a^
5 days	Control	3.80 ± 0.36 ^a^	3.90 ± 0.27 ^a^	20.21 ± 1.39 ^a^	5.00 ± 0.21 ^a^	97.47 ± 5.52 ^a^	4.44 ± 0.48 ^a^
	NaCl	3.45 ± 0.30 ^a^	3.00 ± 0.26 ^b^	19.15 ± 1.94 ^a^	3.90 ± 0.28 ^b^	68.45 ± 4.65 ^b^	3.02 ± 0.16 ^b^
	EBL + NaCl	3.70 ± 0.28 ^a^	3.20 ± 0.15 ^b^	20.80 ± 1.45 ^a^	4.10 ± 0.18 ^b^	86.35 ± 3.25 ^ac^	3.75 ± 0.42 ^a^
	EBL THS + NaCl	4.28 ± 0.36 ^ab^	3.15 ± 0.36 ^ab^	19.30 ± 2.27 ^a^	4.00 ± 0.21 ^b^	94.44 ± 5.31 ^a^	4.09 ± 0.24 ^a^
7 days	Control	4.20 ± 0.51 ^a^	4.25 ± 0.40 ^a^	23.00 ± 0.97 ^a^	5.00 ± 0.21 ^a^	101.98 ± 6.04 ^a^	4.64 ± 0.43 ^a^
	NaCl	3.44 ± 0.36 ^a^	3.05 ± 0.34 ^b^	22.50 ± 2.03 ^a^	4.30 ± 0.31 ^b^	70.45 ± 3.64 ^b^	3.26 ± 0.30 ^b^
	EBL + NaCl	3.75 ± 0.28 ^a^	3.30 ± 0.30 ^b^	20.80 ± 2.41 ^a^	4.20 ± 0.25 ^b^	90.24 ± 6.79 ^a^	4.32 ± 0.47 ^a^
	EBL THS + NaCl	4.80 ± 0.33 ^a^	3.70 ± 0.44 ^ab^	22.65 ± 2.71 ^a^	4.50 ± 0.34 ^ab^	98.13 ± 9.92 ^a^	4.92 ± 0.24 ^a^

The values are presented as the means ± SDs for each treatment. Values not sharing a common or same alphabet letter (a–c), and they differ significantly at *p* < 0.05 (Duncan’s multiple range test).

**Table 2 plants-14-03404-t002:** Effect of 24-epibrassinolide tetrahemisuccinate (EBL THS, 10 nM) and 24-epibrassinolide (EBL, 10 nM) on the content of inorganic ions in leaves, stems, and roots of rapeseed plants (*Brassica napus* L.) under chloride salinization (150 mM NaCl).

**Treatment**	**Leaf**	**Stem**	**Root**	**Leaf**	**Stem**	**Root**	**Leaf**	**Stem**	**Root**
	Na^+^, At %			K^+^, At %			Cl^−^, At %		
Control	1.23 ± 0.09 ^a^	1.74 ± 0.40 ^a^	2.72 ± 0.23 ^a^	48.31 ± 2.16 ^a^	56.84 ± 1.30 ^a^	34.85 ± 2.16 ^a^	1.75 ± 0.27 ^a^	1.48 ± 0.19 ^a^	2.11 ± 0.13 ^a^
NaCl	25.47 ± 1.34 ^b^	24.37 ± 1.95 ^b^	17.25 ± 1.48 ^b^	17.16 ± 2.14 ^b^	29.98 ± 3.36 ^b^	19.45 ± 0.67 ^b^	38.30 ± 1.39 ^b^	25.55 ± 1.88 ^b^	22.69 ± 1.37 ^b^
EBL + NaCl	20.02 ± 0.80 ^c^	18.90 ± 0.95 ^c^	16.99 ± 0.35 ^b^	14.77 ± 0.90 ^b^	31.76 ± 1.20 ^b^	20.08 ± 2.87 ^b^	38.46 ± 2.22 ^b^	24.52 ± 0.68 ^b^	22.86 ± 1.35 ^b^
EBL THS + NaCl	20.79 ± 0.80 ^c^	17.12 ± 1.45 ^c^	16.75 ± 0.84 ^b^	15.12 ± 0.74 ^b^	31.55 ± 3.06 ^b^	23.48 ± 0.60 ^c^	23.45 ± 0.79 ^c^	23.07 ± 1.84 ^b^	38.86 ± 1.60 ^c^
	Mg^++^, At %			Ca^++^, At %			S^++^, At %		
Control	7.52 ± 0.57 ^a^	4.87 ± 0.23 ^a^	4.74 ± 0.67 ^a^	21.67 ± 1.34 ^a^	8.28 ± 0.50 ^a^	16.20 ± 1.78 ^a^	7.65 ± 0.48 ^a^	8.16 ± 0.45 ^a^	9.51 ± 1.02 ^a^
NaCl	3.61 ± 0.24 ^b^	3.09 ± 0.86 ^b^	2.31 ± 0.38 ^b^	5.38 ± 0.46 ^b^	3.06 ± 0.29 ^b^	9.28 ± 1.09 ^b^	3.43 ± 0.29 ^b^	4.54 ± 0.23 ^b^	7.18 ± 0.43 ^b^
EBL + NaCl	5.76 ± 0.26 ^c^	4.26 ± 0.28 ^a^	2.76 ± 0.39 ^b^	9.76 ± 1.11 ^c^	4.54 ± 0.38 ^c^	7.78 ± 0.67 ^b^	3.86 ± 0.48 ^b^	4.79 ± 0.38 ^b^	6.65 ± 0.11 ^b^
EBL THS + NaCl	5.98 ± 0.11 ^c^	4.87 ± 0.40 ^a^	2.73 ± 0.18 ^b^	10.04 ± 0.45 ^c^	5.65 ± 0.24 ^d^	7.74 ± 0.24 ^c^	3.43 ± 0.24 ^b^	4.15 ± 0.15 ^b^	6.68 ± 0.57 ^b^
	P^+++^, At %			Al^+++^, At %			Fe^++^, At %		
Control	7.71 ± 0.72 ^a^	11.31 ± 1.11 ^a^	17.87 ± 1.55 ^a^	1.11 ± 0.06 ^a^	1.00 ± 0.22 ^a^	1.54 ± 0.40 ^a^	7.06 ± 0.64 ^a^	5.82 ± 0.50 ^a^	9.26 ± 0.63 ^a^
NaCl	3.81 ± 0.62 ^b^	5.36 ± 0.28 ^b^	13.42 ± 0.62 ^b^	0.66 ± 0.13 ^b^	0.75 ± 0.02 ^a^	0.63 ± 0.13 ^b^	3.27 ± 0.32 ^b^	3.90 ± 0.29 ^b^	7.81 ± 0.44 ^b^
EBL + NaCl	2.62 ± 0.24 ^c^	6.51 ± 0.28 ^c^	13.26 ± 0.69 ^b^	0.63 ± 0.13 ^b^	0.39 ± 0.05 ^b^	0.79 ± 0.10 ^b^	2.68 ± 0.25 ^b^	4.44 ± 0.32 ^b^	8.66 ± 0.42 ^a^
EBL THS + NaCl	3.03 ± 0.24 ^b^	7.36 ± 0.67 ^c^	13.62 ± 0.15 ^b^	0.75 ± 0.03 ^b^	0.64 ± 0.02 ^c^	0.79 ± 0.03 ^b^	3.36 ± 0.32 ^b^	4.26 ± 0.14 ^b^	8.25 ± 0.83 ^a^

The values are presented as the means ± SDs for each treatment. Values not sharing a common or the same alphabet letter (a–c), and they differ significantly at *p* < 0.05 (Duncan’s multiple range test).

**Table 3 plants-14-03404-t003:** Dynamics of water content in the aboveground and underground parts of rapeseed plants (*Brassica napus* L.) in response to pretreatment (4 h) with 24-epibrassinolide tetrahemisuccinate (EBL THS) and 24-epibrassinolide (EBL) followed by salt stress.

Treatment		Water Content (%)	
		Shoot	Roots
1 days	Control	88.36 ± 0.85 ^a^	90.35 ± 0.72 ^a^
	NaCl	87.75 ± 0.67 ^a^	90.35 ± 0.36 ^a^
	EBL + NaCl	88.58 ± 0.38 ^a^	89.84 ± 0.16 ^a^
	EBL THS + NaCl	88.49 ± 0.26 ^a^	90.84 ± 0.84 ^a^
3 days	Control	88.51 ± 0.60 ^a^	89.01 ± 0.76 ^a^
	NaCl	89.61 ± 0.32 ^a^	90.93 ± 0.76 ^a^
	EBL + NaCl	90.24 ± 0.79 ^a^	90.36 ± 0.65 ^a^
	EBL THS + NaCl	90.35 ± 0.22 ^b^	92.43 ± 0.26 ^b^
5 days	Control	90.17 ± 0.76 ^a^	91.84 ± 0.28 ^a^
	NaCl	88.98 ± 0.70 ^a^	92.67 ± 0.92 ^a^
	EBL + NaCl	87.84 ± 0.93 ^a^	91.14 ± 0.25 ^a^
	EBL THS + NaCl	88.33 ± 0.79 ^a^	87.76 ± 0.79 ^b^
7 days	Control	85.85 ± 0.82 ^a^	90.14 ± 1.40 ^a^
	NaCl	83.88 ± 0.65 ^b^	89.20 ± 0.16 ^a^
	EBL + NaCl	86.72 ± 1.11 ^a^	90.81 ± 0.43 ^a^
	EBL THS + NaCl	85.35 ± 0.72 ^a^	89.57 ± 0.80 ^a^

The values are presented as the means ± SDs for each treatment. Values not sharing a common or the same alphabet letter (a, b) differ significantly at *p* < 0.05 (Duncan’s multiple range test).

**Table 4 plants-14-03404-t004:** Dynamics of the TBARS content in rapeseed plants (*Brassica napus* L.) in response to pretreatment (4 h) with 24-epibrassinolide tetrahemisuccinate (EBL THS) or 24-epibrassinolide (EBL) followed by salt stress.

Treatment		TBARS Content (µM/g Fresh Weight)		
		Leaves	Stem	Roots
1 days	Control	0.026 ± 0.001 ^a^	0.012 ± 0.001 ^a^	0.024 ± 0.002 ^a^
	NaCl	0.037 ± 0.003 ^b^	0.025 ± 0.003 ^b^	0.039 ± 0.003 ^b^
	EBL + NaCl	0.033 ± 0.003 ^b^	0.025 ± 0.002 ^b^	0.039 ± 0.004 ^b^
	EBL THS + NaCl	0.033 ± 0.002 ^b^	0.018 ± 0.001 ^c^	0.024 ± 0.001 ^a^
3 days	Control	0.036 ± 0.002 ^a^	0.012 ± 0.001 ^a^	0.028 ± 0.002 ^a^
	NaCl	0.043 ± 0.003 ^b^	0.021 ± 0.001 ^b^	0.045 ± 0.003 ^b^
	EBL + NaCl	0.039 ± 0.003 ^ab^	0.012 ± 0.001 ^a^	0.026 ± 0.002 ^a^
	EBL THS + NaCl	0.036 ± 0.002 ^a^	0.021 ± 0.002 ^b^	0.035 ± 0.003 ^c^
5 days	Control	0.042 ± 0.004 ^a^	0.012 ± 0.001 ^a^	0.017 ± 0.001 ^a^
	NaCl	0.067 ± 0.006 ^b^	0.016 ± 0.001 ^b^	0.025 ± 0.002 ^b^
	EBL + NaCl	0.042 ± 0.003 ^a^	0.021 ± 0.003 ^b^	0.015 ± 0.002 ^a^
	EBL THS + NaCl	0.041 ± 0.006 ^a^	0.020 ± 0.002 ^b^	0.027 ± 0.003 ^b^
7 days	Control	0.021 ± 0.002 ^a^	0.012 ± 0.001 ^a^	0.015 ± 0.001 ^a^
	NaCl	0.049 ± 0.004 ^b^	0.024 ± 0.002 ^b^	0.021 ± 0.002 ^b^
	EBL + NaCl	0.024 ± 0.003 ^a^	0.022 ± 0.002 ^b^	0.016 ± 0.001 ^a^
	EBL THS + NaCl	0.020 ± 0.005 ^a^	0.025 ± 0.002 ^b^	0.022 ± 0.002 ^b^

The values are presented as the means ± SDs for each treatment. Values not sharing a common or the same alphabet letter (a–c) differ significantly at *p* < 0.05 (Duncan’s multiple range test).

## Data Availability

Data are contained within the article and Appendix A.

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
