# Peer review of "24-Epibrassinolide-Succinic Acid Conjugate Is Involved in the Acclimation of Rape Plants to Salt Stress"

_plants, 2025, doi:10.3390/plants14213404_

Round 1
Reviewer 1 Report
Comments and Suggestions for Authors
The manuscript titled “24-Epibrassinolide–succinic acid conjugate is involved in acclimation of rape plants to salt stress” presents findings that may be of interest to plant researchers. However, the results currently offer only basic insights and lack depth in interpretation.
The methodology section, particularly the description of lipid peroxidation measurement using TBARS, is unclear and requires more detail. Additionally, there is no information provided regarding the use of antioxidants, which is critical for validating the oxidative stress-related findings.
To strengthen the manuscript, I strongly recommend a thorough revision of the methodology to ensure clarity and reproducibility. Including relevant literature to support the findings would also enhance the scientific value of the work. A more comprehensive and detailed methods section will significantly improve the manuscript’s overall quality and credibility.
Author Response
Dear Reviewer
The authors consider it their pleasant duty to sincerely thank the distinguished reviewer for useful recommendations, and friendly criticism.
Kind regards
Liliya Kolomeichuk
Response to reviewer' comments
Comment 1: “The manuscript titled “24-Epibrassinolide–succinic acid conjugate is involved in acclimation of rape plants to salt stress” presents findings that may be of interest to plant researchers. However, the results currently offer only basic insights and lack depth in interpretation.”
Response 1: The authors are grateful for such a significant question. Indeed, this is one of the first studies to elucidate the protective role of EBL THS in plants. Nevertheless, we believe we have demonstrated a fairly broad spectrum of defense responses involving this conjugate under salt stress. In response to your suggestion, we have significantly revised the manuscript's Methodology section and Discussion to provide a more comprehensive interpretation of the obtained results. For this purpose, certain sections of the Discussion section have been rewritten. Furthermore, it is suggested that EBL THS may enhance rapeseed salt tolerance by stimulating the accumulation of proline, which acts not only as a "chemical chaperone" but also as an antioxidant. A discussion of the possible mechanisms by which proline exerts its antioxidant properties is presented.
Comment 2: “The methodology section, particularly the description of lipid peroxidation measurement using TBARS, is unclear and requires more detail. Additionally, there is no information provided regarding the use of antioxidants, which is critical for validating the oxidative stress-related findings.”
Response 2: Thank you very much for this important question. We have carefully redesigned the Materials and Methods section to provide a more detailed and reproducible explanation of the TBARS method for measuring lipid peroxidation. This is a widely used method in experimental plant biology and is widely cited in the scientific literature. In addition, in response to an important point regarding antioxidants, we have added detailed information on the definition of a non-enzymatic antioxidant, such as proline, and discussed the possibility of EBL THS's protective function, which may be mediated through proline's antioxidant role. Furthermore, we presented stability data for the EBL THS conjugate we used (see Methods S1), which is also important for substantiating the validity of our study.
Comment 3: “To strengthen the manuscript, I strongly recommend a thorough revision of the methodology to ensure clarity and reproducibility. Including relevant literature to support the findings would also enhance the scientific value of the work. A more comprehensive and detailed methods section will significantly improve the manuscript’s overall quality and credibility.”
Response 3: Thank you for paying special attention to the methodological part of the manuscript.We have carried out a comprehensive revision of the entire manuscript, paying special attention to the section "Materials and methods" to ensure maximum clarity and reproducibility, as recommended. All missing experimental procedures were carefully described.
In addition, we presented the data in two tables and in the form of graphs to make the material easier to understand, rewrote some fragments of the Discussion section to improve the presentation and enhance the scientific clarity of the material, and significantly edited the English language.
For your information, we submitted the manuscript in two versions:
- Revised final file, in which all changes to the text are highlighted in red and English language edits are highlighted in blue.
- Revised with tracking system. In this file, new additions are highlighted in green, deletions are highlighted in red, and English language edits are highlighted in blue.
We are grateful for the reviewer’s constructive feedback and opportunity to improve our article.
Reviewer 2 Report
Comments and Suggestions for Authors
In this manuscript submitted to Plants, the authors describe the effects of 24-epibrassinolide-succinic acid conjugate on increasing salt stress tolerance in rapeseed plants. However, there are several remarks (methods are in many points incomplete and important information is missing) which prevent my accepting and revisions are required as follows:
I would be interested to know if the stability of the EBL-THS conjugate was investigated. If so, this data should be included in the supplementary information.
Why did the authors perform pretreatment using SA and MM (mechanical mixture: SA + EBL)? This should be explained.
I recommend presenting the data in graphs rather than tables, as these are not very clear.
There are a few minor errors: the abbreviations THS-EBL and TGC-EBL are mixed up and should be unified.
The decimal numbers in Table S1 are incorrect.
Author Response
Dear Reviewer
The authors consider it their pleasant duty to sincerely thank the distinguished reviewer for useful recommendations, and friendly criticism.
Kind regards
Liliya Kolomeichuk
Response to reviewer' comments
Comment 1: “I would be interested to know if the stability of the EBL-THS conjugate was investigated. If so, this data should be included in the supplementary information.”
Response 1: Thank you for this extremely important question. We conducted a stability study of the EBL-THS conjugate in advance. Detailed information is provided in the supplementary materials (Methods S1). The study demonstrated that the conjugate is stable, as judged by its ability to maintain spectral and chromatographic properties over a long shelf life (approximately one year, at -5°C).
Comment 2: “Why did the authors perform pretreatment using SA and MM (mechanical mixture: SA + EBL)? This should be explained.”
Response 2: According to our observations, separate components of an ether (in our case, a steroidal alcohol and an organic acid) are capable of eliciting a physiological effect in both plant and animal cells that is similar to that of the ether itself (in our case, tetrasuccinate). One possible explanation for this is that the ether may undergo hydrolysis catalyzed by cellular enzymes, and its chemical components may act independently as active agents. In such a case, their separate entry into the cell (even considering altered bioavailability) may, to some extent, replicate the effect of the ether itself, which is of practical interest. Объяснение представлено в методической части рукописи.
Comment 3: “I recommend presenting the data in graphs rather than tables, as these are not very clear.”
Response 3: We agree that graphical representation can often enhance the clarity of data presentation. We have replaced two tables into two graphs..
Comment 4: “There are a few minor errors: the abbreviations THS-EBL and TGC-EBL are mixed up and should be unified.”
Response 4: Thanks for the comment. Errors have been fixed
Comment 5: “The decimal numbers in Table S1 are incorrect.”
Response 5: Thanks for the comment. We have corrected all errors in the Table S1.
In addition, we rewrote some fragments of the Discussion section to improve the presentation and enhance the scientific clarity of the material. The methodological part of the manuscript was described in more detail and significantly edited the English language.
For your information, we submitted the manuscript in two versions:
- Revised final file, in which all changes to the text are highlighted in red and English language edits are highlighted in blue.
- Revised with tracking system. In this file, new additions are highlighted in green, deletions are highlighted in red, and English language edits are highlighted in blue.
We are grateful for the reviewer’s constructive feedback and opportunity to improve our article.
Round 2
Reviewer 1 Report
Comments and Suggestions for Authors
Thank you for providing additional information and improved the manuscript, and I am very pleased with the revised version.
Reviewer 2 Report
Comments and Suggestions for Authors
I agree with the proposed revisions and have no further comments.